# Morphological and Phylogenetic Analyses Reveal Three New Species of *Pestalotiopsis* (*Sporocadaceae*, *Amphisphaeriales*) from Hainan, China

**DOI:** 10.3390/microorganisms11071627

**Published:** 2023-06-21

**Authors:** Zhaoxue Zhang, Jie Zhang, Duhua Li, Jiwen Xia, Xiuguo Zhang

**Affiliations:** Shandong Provincial Key Laboratory for Biology of Vegetable Diseases and Insect Pests, College of Plant Protection, Shandong Agricultural University, Taian 271018, China; zhangzhaoxue2022@126.com (Z.Z.); zhjie8087@163.com (J.Z.); ldh3458198584@163.com (D.L.); xiajiwen1@126.com (J.X.)

**Keywords:** *Pestalotiopsis*, morphology, new species, phylogeny

## Abstract

Species of *Pestalotiopsis* were mainly introduced as endophytes, plant pathogens or saprobes from various hosts. In this study, ten strains were isolated from *Ficus macrocarpa*, *Phoebe zhennan* and *Spatholobus suberectus* in China. Based on multilocus phylogenies from the internal transcribed spacer (ITS), the partial translation elongation factor 1-alpha gene (*tef1α*) and the partial beta-tubulin gene (*tub2*), in conjunction with morphological characteristics, we describe three new species, viz., *Pestalotiopsis ficicola* sp. nov., *P. phoebes* sp. nov. and *P. spatholobi* sp. nov.

## 1. Introduction

*Pestalotiopsis* Steyaert, belonging to *Sporocadaceae* (*Amphisphaeriales*, *Ascomycota*), was introduced by Steyaert in 1949 (type species: *Pestalotiopsis guepinii* De Not.) [1]. Species of *Pestalotiopsis* are endophytic, plant pathogenic or saprobic and are associated with a wide range of host plants [2,3,4,5,6]. Currently, a total of 402 names are documented for *Pestalotiopsis* in the Index Fungorum (http://www.indexfungorum.org/, accessed on 16 May 2023). Initially, *Pestalotiopsis* resembling those taxa having affinities with *Pestalotia* were also known as pestalotioid fungi. Pestalotioid fungi are easily characterized by multiseptate and more or less fusiform conidia with appendages at one or both ends, frequently with some melanized cells.

*Pestalotia*, isolated from *Vitis* sp., was introduced by De Not. in 1841 [7]. Subsequently, Steyaert split *Pestalotia* into three genera, viz., *Pestalotia* (with six-celled conidia), *Pestalotiopsis* (with five-celled conidia) and *Truncatella* (with four-celled conidia), based on the number of cells [1]. Although Guba [8,9] initiated some controversy, Steyaert [10,11,12] provided further evidence in support of splitting *Pestalotia*. Sutton [13] accepted most of the genera discussed here (*Pestalotia*, *Pestalotiopsis* and *Truncatella*) which fit into fairly well-defined groups and cited the electron microscope investigation of Griffiths and Swart [14] to support Steyaert’s division of *Pestalotiopsis*. Molecular phylogenetic analysis largely promoted the development of taxonomy. Jeewon et al. [15] established a phylogenetic tree by analyzing some ITS sequences of *Pestalotiopsis* species and found that *Pestalotiopsis* species could be divided into three large branches, with the color type of colored cells and the morphology of the end of the apical accessory filaments as the main distinguishing characteristics. Hu et al. analyzed the ITS and tubulin gene of *Pestalotiopsis*, combined with morphological characteristics and molecular data, and analyzed the relationship between species of *Pestalotiopsis* [16]. Maharachch. et al. split *Pestalotiopsis sensu lato* into three genera—*Pestalotiopsis sensu stricto*, *Neopestalotiopsis* and *Pseudopestalotiopsis*—based on phylogeny of multiple genes and conidial morphology; *Pseudopestalotiopsis* and *Pestalotiopsis* can be easily distinguished from *Neopestalotiopsis* by its versicolorous median cells [6]. Liu et al. included *Pestalotiopsis* into *Sporocadaceae* using morphological characteristics and on the basis of a multilocus phylogenetic analysis [5]. Some genera of the *Sporocadaceae* can be divided into three categories based on the number of appendages, viz., genera with a single apical and basal appendage (*Monochaetia*, *Seiridium*), other genera which do not form appendages (*Nonappendiculata*) or genera which have 2–4 appendages (*Pestalotiopsis*, *Ciliochorella*, *Neopestalotiopsis* and *Pseudopestalotiopsis*) [17]. Recently, Jiang et al. obtained 43 isolates of *Pestalotiopsis* from diseased leaf tissues of Fagaceae based on combined morphology and phylogeny between 2016 and 2021 [18].

In this study, we produced a collection of the genera *Pestalotiopsis* species from leaves of *Ficus macrocarpa*, *Phoebe zhennan* and *Spatholobus suberectus* in East Harbour National Nature Reserve, Hainan Province, China. Three new species were described based on unique morphological characters and distinct phylogenetic placement, viz., *Pestalotiopsis ficicola* sp. nov., *P. phoebes* sp. nov. and *P. spatholobi* sp. nov.

## 2. Materials and Methods

### 2.1. Isolation and Morphology

Samples of *Ficus macrocarpa* L. f., *Phoebe zhennan* S. Lee et F. N. Wei and *Spatholobus suberectus* Dunn showing obvious disease spots were collected from Hainan Province during 2021 in East Harbour National Nature Reserve (110°32′~110°37′ E, 19°51′~20°1′ N), China. The cultures of *Pestalotiopsis* were isolated from diseased tissues of the sample leaves using tissue isolation methods [19]. Fragments (5 × 5 mm) were taken from the edges of the leaf lesions, surface sterilized for 30 s in 75% ethanol, rinsed in sterile deionized water for 30 s, rinsed in 5% sodium hypochlorite solution for 1 min and then rinsed four times in sterile deionized water for 30 s [17]. The pieces were blotted on sterile filter paper to dry, transferred onto PDA flats (PDA medium: potato 200 g, agar 15–20 g, dextrose 15–20 g, deionized water 1 L, pH~7.0, available after sterilization) and incubated at 25 °C for 2–3 days. The edges of hyphal growth were then transferred to new PDA flats to obtain pure cultures; simultaneously, they were inoculated on PDA and incubated at 23 °C under continuous near-ultraviolet light to promote sporulation [20].

All *Pestalotiopsis* plates were incubated at 25 °C for 14 days and morphological characters including graphs of the colonies were documented on the 14th day using a digital camera (Canon G7X, Canon, Tokyo, Japan). Morphological characters of conidiomata were studied using a stereomicroscope (Olympus SZX10, Olympus Corporation, Tokyo, Japan) while the micromorphological structures were observed using a microscope (Olympus BX53, Olympus Corporation, Tokyo, Japan). All cultures were deposited in 10% sterilized glycerin and sterile water at 4 °C for future studies. Micromorphological structural measurements were taken using the Digimizer software v. 5.6.0 (https://www.digimizer.com/, accessed on 16 May 2023), with 20 measurements taken for each structure [17]. Voucher specimens were deposited in the Herbarium Mycologicum Academiae Sinicae, Institute of Microbiology, Chinese Academy of Sciences, Beijing, China (HMAS) and the Herbarium of the Department of Plant Pathology, Shandong Agricultural University, Taian, China (HSAUP). Ex-holotype living cultures were deposited in the Shandong Agricultural University Culture Collection (SAUCC). Taxonomic information of the new taxa was submitted to MycoBank (http://www.mycobank.org, accessed on 16 May 2023).

### 2.2. DNA Extraction and Amplification

Genomic DNA was extracted from the colonies grown on PDA using a kit method (OGPLF-400, GeneOnBio Corporation, Changchun, China) [21]. Gene sequences were obtained from five loci including the internal transcribed spacer regions with the intervening 5.8S nrRNA gene (ITS), the partial translation elongation factor 1-alpha gene (*tef1α*) and the partial beta-tubulin gene (*tub2*). These were amplified by the primer pairs and polymerase chain reaction (PCR) programs listed in Table 1. Amplification reactions were performed in a 25 μL reaction volume which contained 10 μL 2 × Hieff Canace^®^ Plus PCR Master Mix (With Dye) (Yeasen Biotechnology, Cat No. 10154ES03, Shanghai, China), 0.5 μL of each forward and reverse primer (10 μM) (TsingKe, Qingdao, China) and 1 μL template genomic DNA, adjusted with distilled deionized water to a total volume of 25 μL. PCR amplification products were visualized on 2% agarose electrophoresis gel. DNA sequencing was performed using an Eppendorf Master Thermocycler (Hamburg, Germany) at the TsingKe Company Limited (Qingdao, China) bidirectionally. Consensus sequences were obtained using MEGA 7.0 [22]. All sequences generated in this study were deposited in GenBank (Appendix A: See Appendix A).

### 2.3. DNA Extraction and Amplification

Novel sequences obtained in this study and related sets of sequences from Jiang et al. [18] were aligned with MAFFT v. 7 and corrected manually using MEGA 7 [27]. Multilocus phylogenetic analyses were based on the algorithms maximum likelihood (ML) and Bayesian inference (BI) methods. The ML was run on the CIPRES Science Gateway portal (https://www.phylo.org, accessed on 16 May 2023) [28] using RAxML–HPC2 on XSEDE v. 8.2.12 [29] and employed a GTRGAMMA substitution model with 1000 bootstrap replicates. Other parameters were default. For Bayesian inference analyses, the best model of evolution for each partition was determined using ModelTest v. 2.3 [30] and included the analyses. The BI was performed using MrBayes on XSEDE v. 3.2.7a [31,32,33] and two Markov chain Monte Carlo (MCMC) chains were run, starting from random trees, for 2,000,000 generations (average standard deviation of split frequencies < 0.01). Additionally, a sampling frequency of 100 generations was used. The first 25% of trees were discarded as burn-in and BI posterior probabilities (PPs) were conducted from the remaining trees. The consensus trees were optimized using FigTree v. 1.4.4 (http://tree.bio.ed.ac.uk/software/figtree, accessed on 16 May 2023) and embellished with Adobe Illustrator CC 2019 (Figure 1).

## 3. Results

### 3.1. Phylogenetic Analyses

The alignment contained 182 strains representing *Pestalotiopsis* and the strain MFLUCC 12-0652 of *Neopestalotiopsis magna* was used as outgroup [6]. The dataset had an aligned length of 2011 characters including gaps, viz., ITS: 1–565, *tef1α*: 566–1149 and *tub2*: 1150–2011 (Appendix A). Of these, 1172 were constant, 281 were parsimony uninformative and 558 were parsimony informative. The ModelTest suggested that the BI used the Dirichlet base frequencies, the GTR + I + G evolutionary mode for ITS and *tub2*, and HKY + I + G for *tef1α*.

The topology of the ML tree was consistent with that of the Bayesian tree; therefore, it only showed the topology of the ML tree as a representative for summarizing the evolutionary relationship within the genus *Pestalotiopsis*. The final ML optimization likelihood was −17,300.001718. The 181 strains were assigned to 109 species clades on the phylogram (Figure 1). Based on the phylogenetic resolution and morphological analyses, the present study introduced three novel species of the *Pestalotiopsis*, viz., *Pestalotiopsis ficicola* sp. nov., *P. phoebes* sp. nov. and *P. spatholobi* sp. nov.

### 3.2. Taxonomy

#### 3.2.1. *Pestalotiopsis ficicola* Z.X. Zhang, J.W. Xia and X.G. Zhang, sp. nov.

MycoBank: No. MB848628

Etymology: The epithet “*ficicola*” pertains to the generic name of the host plant *Ficus microcarpa*.

Type: China, Hainan Province, East Harbour National Nature Reserve, on diseased leaves of *Ficus microcarpa*, 23 May 2021, Z.X. Zhang, holotype HMAS 352477, ex-holotype living culture SAUCC230046.

Description: Conidiomata in culture sporodochial, aggregated or solitary, erumpent, pulvinate, straw yellow, exuding black conidial masses. Conidiophores simple, hyaline, subcylindrical, usually reduced to conidiogenous cells. Conidiogenous cells aggregative, hyaline, smooth, cylindrical, 16.5–22.4 × 4.1–6.0 μm. Conidia fusoid, straight or slightly curved, 4-septate, smooth, slightly constricted at the septa, 18.1–22.7 × 5.6–7.9 μm; basal cell obconic with a truncate base, thin-walled, hyaline, 3.6–5.2 μm; median cells 3, trapezoidal or subcylindrical, concolorous, pale brown to brown, thick-walled, the first median cell from base 3.6–4.3 μm long, the second cell 3.4–5.7 μm long, the third cell 3.5–5.6 μm long, together 11.8–14.2 μm long; apical cell conic with an acute apex, thin-walled, hyaline, 2.0–4.0 μm long; basal appendage single, unbranched, tubular, centric, straight or slightly bent, 3.0–6.7 μm long; apical appendages 2–3, unbranched, tubular, straight or slightly bent, 10.8–25.5 μm long. Sexual morph unknown, see Figure 2.

Culture characteristics: The colonies diameter reached 90 mm after 14 days of dark culture at 25 °C on PDA, flat, aerial mycelium flocculent fluffy, white; reverse center pale yellow, edge white.

Additional specimen examined: China, Hainan Province, East Harbour National Nature Reserve, on diseased leaves of *Ficus microcarpa*, 23 May 2021, Z.X. Zhang, HSAUP230043, living culture SAUCC230043, on diseased leaves of *Ficus microcarpa*, 23 May 2021, Z.X. Zhang, HSAUP230042, living culture SAUCC230042.

Notes: Phylogenetic analyses of three combined genes (ITS, *tef1α* and *tub2*) showed *Pestalotiopsis ficicola* sp. nov. was closely related to *P. formosana* and *P. nanningensis*. In detail, *P. ficicola* was distinguished from *P. formosana* by 7/540 bp in ITS, 3/478 bp in *tef1α* and 3/388 bp in *tub2* and from *P. nanningensis* by 5/540 in ITS, 4/478 in *tef1α* and 8/440 in *tub2*. In morphology, the conidia of *P. ficicola* was longer than *P. formosana* (18.1–22.7 × 5.6–7.9 vs. 8–22 × 6–7 µm) and shorter than *P. nanningensis* (18.1–22.7 × 5.6–7.9 vs. 24–26.5 × 7–8 µm). What is more, the apical appendages of *P. ficicola* were longer than *P. formosana* (18–22.5 vs. 8–20 µm) and shorter than *P. nanningensis* (10.8–25.5 vs. 13.5–26.5 µm) [34,35].

#### 3.2.2. *Pestalotiopsis phoebes* Z.X. Zhang, J.W. Xia and X.G. Zhang, sp. nov.

MycoBank: No. MB848629

Etymology: The epithet “*phoebes*” pertains to the generic name of the host plant *Phoebe zhennan*.

Type: China. Hainan Province, East Harbour National Nature Reserve, on diseased leaves of *Phoebe zhennan*, 23 May 2021, Z.X. Zhang, holotype HMAS 352478, ex-holotype living culture SAUCC230093.

Description: Conidiomata in culture sporodochial, aggregated or solitary, erumpent, pulvinate, black, exuding black conidial masses. Conidiophores simple, hyaline, usually reduced to conidiogenous cells. Conidiogenous cells aggregative, hyaline, smooth, cylindrical, 16.4–36.6 × 3.1–5.3 μm. Conidia fusoid, straight or slightly curved, 4-septate, smooth, slightly constricted at the septa, 20.2–23.5 × 6.4–8.6 μm; basal cell obconic with a truncate base, thin-walled, hyaline, 2.9–4.3 μm; median cells 3, trapezoidal or subcylindrical, concolorous, pale brown to brown, thick-walled, the first median cell from base 4.5–6.1 μm long, the second cell 5–5.9 μm long, the third cell 3.5–4.9 μm long, together 13.7–15.7 μm long; apical cell conic with an acute apex, thin-walled, hyaline, 2.9–4.1 μm long; basal appendage single, unbranched, tubular, centric, straight or slightly bent, 4.7–6.2 μm long; apical appendages 2–4, unbranched, tubular, straight or slightly bent, 15.2–19.6 μm long. Sexual morph unknown, see Figure 3.

Culture characteristics: The colonies diameter reached 90 mm after 14 days of dark culture at 25 °C on PDA, flat, aerial mycelium flocculent fluffy, grows irregularly, white; reverse center had a light brown ring, edge white.

Additional specimen examined: China, Hainan Province, East Harbour National Nature Reserve, on diseased leaves of *Phoebe zhennan*, 23 May 2021, Z.X. Zhang, HSAUP230092, living culture SAUCC230092, on diseased leaves of *Phoebe zhennan*, 23 May 2021, Z.X. Zhang, HSAUP230094, living culture SAUCC230094.

Notes: Phylogenetic analyses of three combined genes (ITS, *tef1α* and *tub2*) showed *Pestalotiopsis phoebes* sp. nov. was closely related to *P. clavata* and *P. pini*. In detail, *P. phoebes* was distinguished from *P. clavata* by 2/540 bp in ITS, 7/259 bp in *tef1α* and 11/441 bp in *tub2* and from *P. pini* by 3/540 in ITS, 9/254 in *tef1α* and 10/762 in *tub2*. In morphology, the conidia of *P. phoebes* was shorter than *P. clavate* (20.2–23.5 × 6.4–8.6 vs. 20–27 × 6.5–8 µm) and *P. pini* (20.2–23.5 × 6.4–8.6 vs. 20.0–27.6 × 4.7–8.2 µm). What is more, the apical appendages of *P. phoebes* were shorter than *P. clavate* (15.2–19.6 vs. 20–25 µm) and *P. pini* (15.2–19.6 vs. 9.7–27.8 µm) [36,37].

#### 3.2.3. *Pestalotiopsis spatholobi* Z.X. Zhang, J.W. Xia and X.G. Zhang, sp. nov.

MycoBank: No. MB848630

Etymology: The epithet “*spatholobi*” pertains to the generic name of the host plant *Spatholobus suberectus*.

Type: China. Hainan Province, East Harbour National Nature Reserve, on diseased leaves of *Spatholobus suberectus*, 23 May 2021, Z.X. Zhang, holotype HMAS 352479, ex-holotype living culture SAUCC231201.

Description: Conidiomata in culture sporodochial, aggregated or solitary, erumpent, exuding black conidial masses, covered with mycelium. Conidiophores simple, hyaline, usually reduced to conidiogenous cells. Conidiogenous cells aggregative, hyaline, smooth, cylindrical, 11.4–26.8 × 2.3–6.2 μm. Conidia fusoid, straight or slightly curved, 4-septate, smooth, slightly constricted at the septa, 16.7–23.0 × 5.9–7.4 μm; basal cell obconic with a truncate base, thin-walled, hyaline, 3.0–5.4 μm; median cells 3, trapezoidal or subcylindrical, concolorous, pale brown to brown, thick-walled, the first median cell from base 3.5–5.0 μm long, the second cell 3.7–5.5 μm long, the third cell 3.6–4.9 μm long, together 11.2–14.9 μm long; apical cell conic with an acute apex, thin-walled, hyaline, 3.1–5.0 μm long; basal appendage single, unbranched, tubular, centric, straight or slightly bent, 0.9–3.1 μm long; apical appendages 1–3, unbranched, tubular, straight or slightly bent, 8.4–15.3 μm long. Sexual morph unknown, see Figure 4.

Culture characteristics: The colonies diameter reached 90 mm after 14 days of dark culture at 25 °C on PDA, flat, aerial mycelium flocculent fluffy, radial stripes from the middle to the periphery, white; reverse center was black, edge white.

Additional specimen examined: China, Hainan Province, East Harbour National Nature Reserve, on diseased leaves of *Spatholobus suberectus*, 23 May 2021, Z.X. Zhang, HSAUP231203, living culture SAUCC231203, on diseased leaves of *Spatholobus suberectus*, 23 May 2021, Z.X. Zhang, HSAUP231204, living culture SAUCC231204, on diseased leaves of *Spatholobus suberectus*, 23 May 2021, Z.X. Zhang, HSAUP231213, living culture SAUCC231213.

Notes: Phylogenetic analyses of three combined genes (ITS, *tef1α* and *tub2*) showed *Pestalotiopsis spatholobi* sp. nov. formed an independent clade and was closely related to *P. diploclisia* and *P. humicola*. In detail, *P. spatholobi* was distinguished from *P. diploclisia* by 2/539 bp in ITS, 11/258 bp in *tef1α* and 12/762 bp in *tub2* and from *P. humicola* by 3/539 in ITS, 10/260 in *tef1α* and 15/762 in *tub2*. In morphology, the conidia of *P. spatholobi* was shorter than *P. diploclisia* (16.7–23.0 × 5.9–7.4 vs. 20–28 × 5–7 µm) and *P. humicola* (16.7–23.0 × 5.9–7.4 vs. 17–23 × 5–7.5 µm). What is more, the apical appendages of *P. spatholobi* were shorter than *P. diploclisia* (8.4–15.3 vs. 10–22 µm) and *P. humicola* (8.4–15.3 vs. 6–13 µm) [6].

## 4. Discussion

In the present study, ten strains from three host genera (*Ficus macrocarpa*, *Phoebe zhennan* and *Spatholobus suberectus*) were split into three new species (*Pestalotiopsis ficicola* sp. nov., *P. phoebes* sp. nov. and *P. spatholobi* sp. nov.) based on phylogeny and morphology. The Global Biodiversity Information Facility (https://www.gbif.org/, accessed on 16 May 2023) contains 4174 georeferenced records of *Pestalotiopsis* species reported around the world. Most species are distributed in countries such as the United States of American, Europe, Australia and China where suitable climates and the environment favors growth of unusual microbial species [38].

Historically, traditional identification of *Pestalotiopsis* species has long been a complicated endeavor [8,9,10,11,12]. Fortunately, with the development of molecular technology and the efforts of many taxonomists, the taxonomic status of *Pestalotiopsis* (*Sporocadaceae*, *Amphisphaeriales*) has become increasingly apparent [5,6,13,18,19]. Recent research showed that the same *Pestalotiopsis* species can be found on hosts belonging to multiple plant families, for example, *P. chamaeropis* was isolated from *Chamaerops humilis* (*Arecaceae*), *Quercus* sp., *Castanopsis* sp. (*Fagaceae*) and *Camellia* sp. (*Theaceae*) [6,18,39]. On the other hand, a variety of *Pestalotiopsis* species were isolated from the same plant host, for example, *Pestalotiopsis chamaeropis*, *P. kenyana*, *P. nanjingensis* and *P. rhodomyrtus* were isolated from the *Quercus aliena* (*Fagaceae*) [18]. These findings indicate that *Pestalotiopsis* species have the ability to infect a wide range of host plants rather than showing restricted host preferences.

In the pestalotioid species, apical or basal appendages differ in number, origin, position, number of branches and branching pattern, and these characters can be divided into some genera of the *Sporocadaceae* [5,17]. *Monochaetia* and *Seiridium* possess single apical and basal appendages, *Nonappendiculata* do not possess any appendages, and *Pestalotiopsis*, *Ciliochorella*, *Neopestalotiopsis*, *Pseudopestalotiopsis* possess 2–4 appendages. In previous studies, some members of *Sporocadaceae* were shown to contain secondary metabolites with significant biological activity, e.g., the sterol constituents of the marine fungus *Pestalotiopsis* sp. XWS03F09 exhibit selective inhibitory activities against antitumor cells [40,41,42,43]. Therefore, it is necessary to carry out genome sequencing of *Pestalotiopsis* species and fully explore, through structural and functional annotation, the diversity of secondary metabolites which might be useful for various biotechnological applications. This shows the importance of taxonomy and deposition of specimens.

## Figures and Tables

**Figure 1 microorganisms-11-01627-f001:**
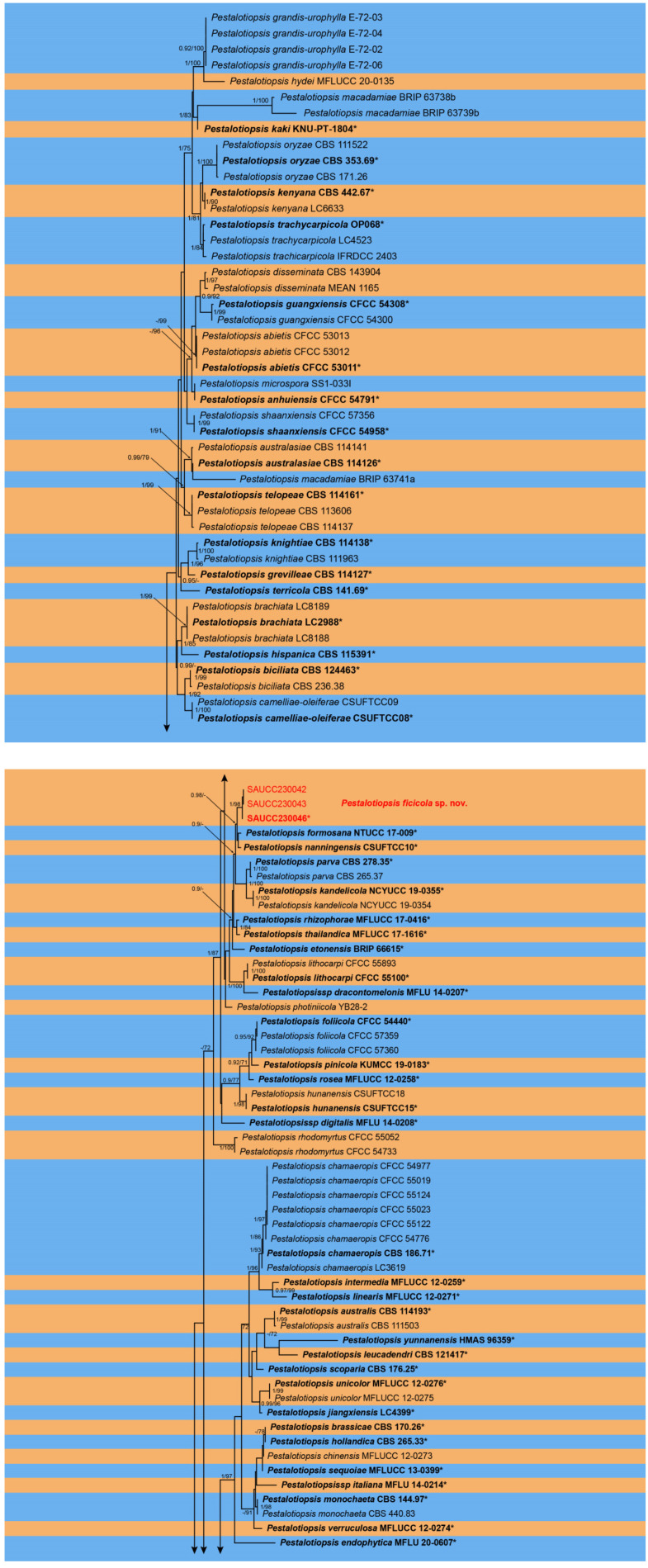
A phylogram of the *Pestalotiopsis* with *Neopestalotiopsis magna* (MFLUCC 12-0652) as outgroup based on a concatenated ITS, *tef1α* and *tub2* sequence alignment (only shows the topology of the ML tree). BI posterior probabilities and maximum likelihood bootstrap support values above 0.70 and 90% are shown at the first and second position, respectively. Ex-type cultures are marked in bold face and *. Strains obtained in the present study are in red. Some branches are shortened for layout purposes—these are indicated by two diagonal lines with the number of times. The orange and blue areas are used to distinguish different species. The scale bar at the left–bottom represents 0.5 substitutions per site.

**Figure 2 microorganisms-11-01627-f002:**
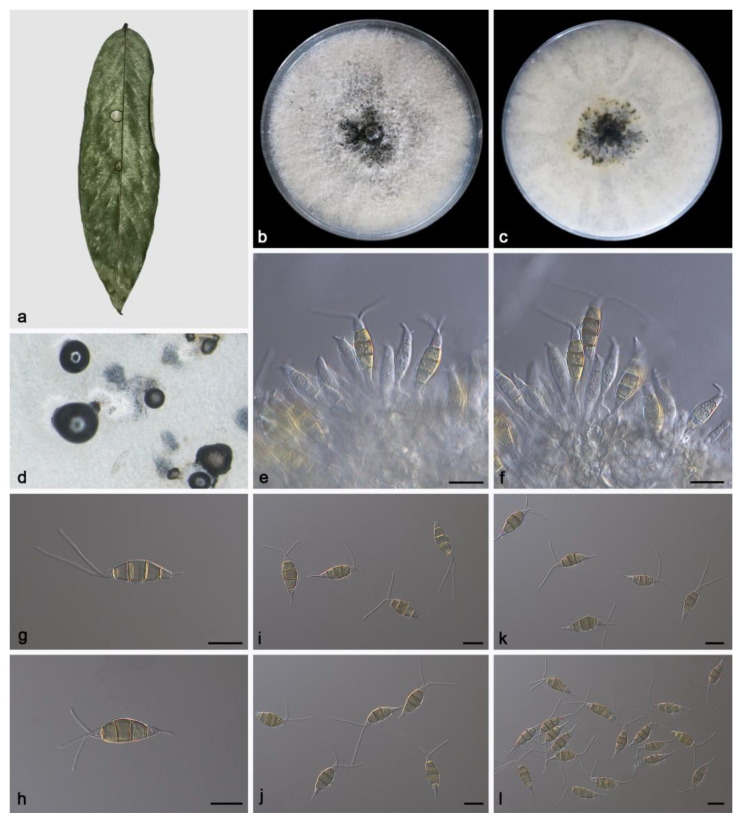
*Pestalotiopsis ficicola* (holotype HMAS 352477). (**a**) Leaf of *Ficus microcarpa*; (**b**,**c**) inverse and reverse sides of colony after 15 days on PDA; (**d**) colony overview; (**e**,**f**) conidiogenous cells with conidia; (**g**–**l**) conidia. Scale bars: (**e**–**l**) 10 μm.

**Figure 3 microorganisms-11-01627-f003:**
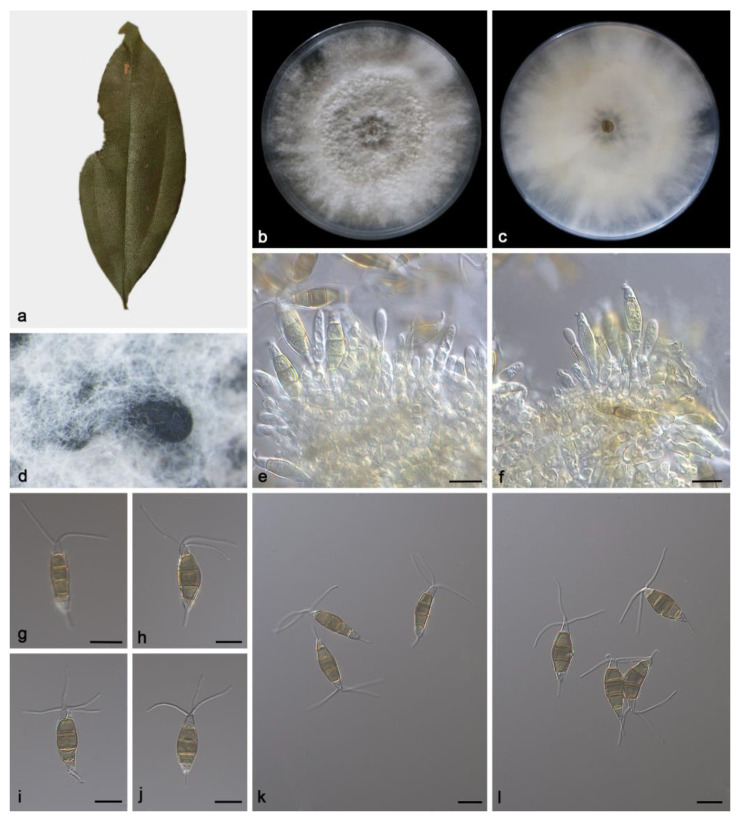
*Pestalotiopsis phoebes* (holotype HMAS 352478). (**a**) Leaf of *Phoebe zhennan*; (**b**,**c**) inverse and reverse sides of colony after 15 days on PDA; (**d**) colony overview; (**e**,**f**) conidiogenous cells with conidia; (**g**–**l**) conidia. Scale bars: (**e**–**l**) 10 μm.

**Figure 4 microorganisms-11-01627-f004:**
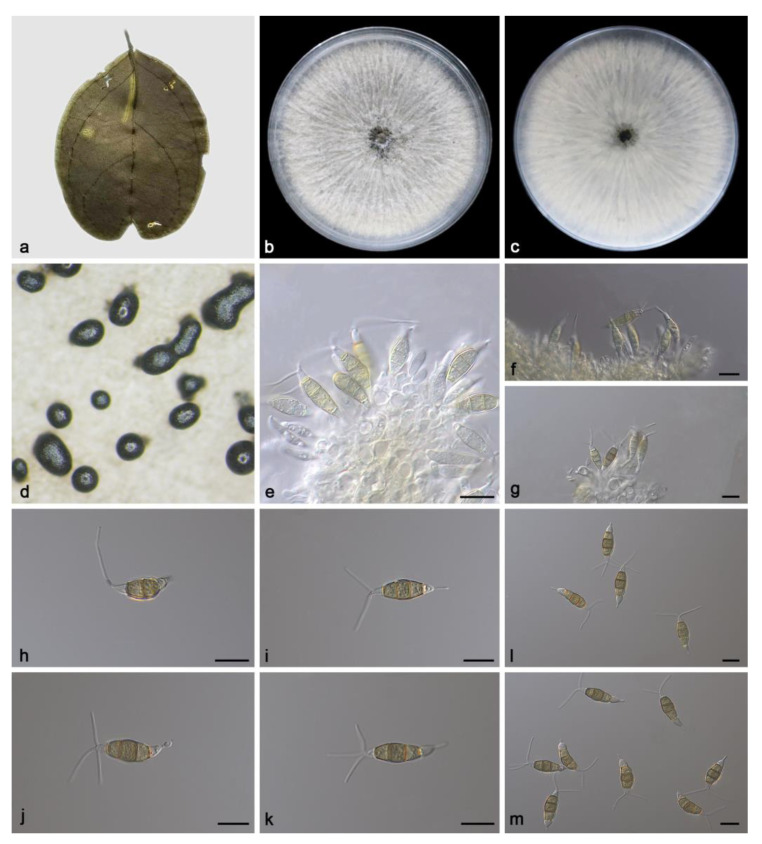
*Pestalotiopsis spatholobi* (holotype HMAS 352479). (**a**) Leaf of *Spatholobus suberectus*; (**b**,**c**) inverse and reverse sides of colony after 15 days on PDA; (**d**) colony overview; (**e**–**g**) conidiogenous cells with conidia; (**h**–**m**) conidia. Scale bars: (**e**–**m**) 10 μm.

**Table 1 microorganisms-11-01627-t001:** Molecular markers and their PCR primers and programs used in this study.

Loci	PCR Primers	Sequence (5′—3′)	PCR Cycles	References
ITS	ITS5 ITS4	GGA AGT AAA AGT CGT AAC AAG G TCC TCC GCT TAT TGA TAT GC	(95 °C: 30 s, 55 °C: 30 s, 72 °C: 1 min) × 35 cycles	[23]
* tef1α *	EF1-728F EF-2	CAT CGA GAA GTT CGA GAA GG GGA RGT ACC AGT SAT CAT GTT	(95 °C: 30 s, 48 °C: 30 s, 72 °C: 1 min) × 35 cycles	[24,25]
* tub2 *	Bt-2a Bt-2b	GGT AAC CAA ATC GGT GCT GCT TTC ACC CTC AGT GTA GTG ACC CTT GGC	(95 °C: 30 s, 53 °C: 30 s, 72 °C: 1 min) × 35 cycles	[26]

## Data Availability

The sequences from the present study were submitted to the NCBI database (https://www.ncbi.nlm.nih.gov/, accessed on 16 May 2023) and the accession numbers were listed in Appendix A.

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
