# Peer review of "Morphological and Phylogenetic Analyses Reveal Three New Species of Pestalotiopsis (Sporocadaceae, Amphisphaeriales) from Hainan, China"

_microorganisms, 2023, doi:10.3390/microorganisms11071627_

Round 1
Reviewer 1 Report
The manuscript entitled „Morphological and phylogenetic analyses reveal three new species of Pestalotiopsis (Sporocadaceae, Amphisphaeriales) from Hainan, China“ describes three new species of Pestalotiopsis genus. Authors sequenced ITS, translation elongation factor and beta-tubulin genes usually used in fungi taxonomy. Phylogenetic analysis showed well supported clades for new deasribed species. Introduction in the manuscript contains information necessary for justification of undertaken study. The material and methods used are described in detail. The obtained results are clearly presented. Discussion is well written.
Author Response
Dear sir/madam,
Thank you for your letter and comments concerning our manuscript “Morphological and phylogenetic analyses reveal three new species of Pestalotiopsis (Sporocadaceae, Amphisphaeriales) from Hainan, China” (microorganisms-2427886).
Your comments are highly insightful and help us greatly improve the quality of our manuscript. We hope that the revisions and our responses as in the postscript would be sufficient to make our manuscript suitable for publication in Microorganisms.
Sincerely yours,
Zhaoxue Zhang
Reviewer 2 Report
The MS of Zheng et al, entitled “Morphological and phylogenetic analyses reveal three new species of Pestalotiopsis (Sporocadaceae, Amphisphaeriales) from Hainan, China” contributes to the documenting of the diversity of Pestalotiopsis on different plant hosts. The data was generated as required and the paper is generally well written, except for the discussion section which needs some serios revision.
Suggestions for improving the MS:
Line 24
Pestalotiopsis also known as pestalotioid fungi. Add a “were” between “Pestalotiopsis” and “known”
Line 31
“Although this has caused controversy for Guba [8–9], and Steyaert”
Should be revised, maybe like “Although Guba [8–9] initiated some controversy, Steyaert …”
Line 37
Remove “the”. Analyzing some ITS …
Line 69-70
This sentence needs revisions: “The edge of hyphal was then removed to new PDA flats to gain pure cultures, simultaneously, inoculate on PNA and incubated at ….”
It could be rewritten something like:
“The edge of hyphal was then transferred to new PDA flats to obtain pure cultures; simultaneously, they were inoculated on PDA and incubated at ….”
Double check is PNA was supposed to be PDA.
Lines 72-76
This sentence has to be revised and split in two.
“All Pestalotiopsis plates were incubated at 25 °C for 14 days, morphological characters should be recorded, including graphs of the colonies were taken at the 14th day using a digital camera (Canon G7X), morphological characters of conidiomata using a stereomicroscope (Olympus SZX10), and micromorphological structures were observed using a microscope (Olympus BX53).”
It could be rewritten something like:
“All Pestalotiopsis plates were incubated at 25 °C for 14 days, morphological characters, including graphs of the colonies were documented on the 14th day using a digital camera (Canon G7X). Morphological characters of conidiomata were studied using a stereomicroscope (Olympus SZX10), while the micromorphological structures were observed using a microscope (Olympus BX53).”
Line 102.
Table 2 should be moved in supplementary data.
Figure 1
Indicate that the phylogram was generated by using ML. This is mentioned in line 141 but it should be in the legend for Fig. 1 as well.
Line 142
Change “recapitulating” with “summarizing”
Line 154
Macrocarpa should be microcarpa.
The discussion section needs serious improvement. The following are suggested.
Line 269
Use either “In the present study” or “in this study”
Line 271
Morphology should be morphology
Lines 272-273
This sentence needs revisions:
“Around the world, the Pestalotiopsis species were reported 4,174 georeferenced records in GBIF (https://www.gbif.org/, accessed on 16 May 2023).
The revised sentence might be something like:
“The Global Biodiversity Information Facility (https://www.gbif.org/, accessed on 16 May 2023) contains 4,174 georeferenced records of Pestalotiopsis species reported around the world.”
Lines 279-286
This section is very poorly written. It has to be revised.
“In recently research, same Pestalotiopsis species can be found on multiple hosts of families, example, P. chamaeropis can be isolated from the host of Arecaceae (Chamaerops humilis), Fagaceae (Quercus sp. & Castanopsis sp.), and Theaceae (Camellia sp.) [6,18,39]. Furthermore, a variety of Pestalotiopsis species can also be isolated from one plant host, example, Pestalotiopsis chamaeropis, P. kenyana, P. nanjingensis and P. rhodomyrtus can be isolated from the host of Fagaceae (Quercus aliena) [18]. This indicated that the Pestalotiopsis species has a wide range of host plant, and hasn’t possessed host preferences.
It is always advisable to write first the species name and then the family. The revised section could look something like:
““Recent research showed that the same Pestalotiopsis species can be found on hosts belonging to multiple plant families; for example, P. chamaeropis was isolated from Chamaerops humilis (Arecaceae), Quercus sp. & Castanopsis sp. (Fagaceae), and Camellia sp. (Theaceae) [6,18,39]. On the other hand, a variety of Pestalotiopsis species were isolated from the same plant host; for example, Pestalotiopsis chamaeropis, P. kenyana, P. nanjingensis and P. rhodomyrtus were isolated from the Quercus aliena (Fagaceae) [18]. These findings indicate that Pestalotiopsis species have the ability to infect a wide range of host plants rather than showing restricted host preferences.”
Lines 289-297
Same situation in this section. Please revise.
“Monochaetia and Seiridium has possessed single apical and basal appendage, Nonappendiculata hasn’t possessed appendages, and Pestalotiopsis, Ciliochorella, Neopestalotiopsis, Pseudopestalotiopsis has possessed 2–4 appendages. In the previous studies, some members of Sporocadaceae contains secondary metabolites with significant biological activity, e.g. the sterol constituents of marine fungus Pestalotiopsis sp. XWS03F09 exhibited selective inhibitor activities against anti-tumor cells [40–43]. Therefore, it is necessary carry out genome sequencing of Pestalotiopsis species, and fully explore secondary metabolites with abundant activity through structural and functional annotation.”
The revised section could look something like:
“Monochaetia and Seiridium possess single apical and basal appendage, Nonappendiculata doesn’t possess appendages, and Pestalotiopsis, Ciliochorella, Neopestalotiopsis, Pseudopestalotiopsis possess 2–4 appendages. In previous studies, some members of Sporocadaceae were shown to contain secondary metabolites with significant biological activity, e.g., the sterol constituents of the marine fungus Pestalotiopsis sp. XWS03F09 exhibit selective inhibitory activities against anti-tumor cells [40–43]. Therefore, it is necessary to carry out genome sequencing of Pestalotiopsis species, and fully explore, through structural and functional annotation, the diversity of secondary metabolites which might be useful for various biotechnological applications.”
English language usage needs to be revised in section 2.1 and in section 4. Dicussion. Section 4 looks particularly bad. Some options are provided in the suggestions to authors.
Author Response

(The authors gave the same response as above.)

Reviewer 3 Report
This paper introducing three new species of Pestalotiopsis contains high-quality systematics. My comments are minor - we need to know more about the location and context of the host plants, and there are some issues about the phylogenetic inference methods. The language is generally very readable, but there are a few grammatical issues throughout the manuscript.
11-12 not a full sentence
13 spacers -> spacer
23-25 sentence meaning unclear
33 fitted -> fit
47-50 need to reorder this sentence so the "based on the number of appendages" doesn't fall at the end
51-53 how many species did Jiang et al. find? Why is this study mentioned?
54 established genera? need to rephrase
56 the way this is phrased "combining phylogeny and morphology" makes it seem as if these species were not morphologically distinct. It would be better to say "based on unique morphological characters and distinct phylogenetic placement" if that is indeed the case
54-57 We need more information about these hosts - are these native to Hainan Province? Were they growing in forests, horticultural settings or agriculture?
60-62 need a table or some list of the geographic location of collections
60 were the plants symptomatic or were there actual signs (sporulation) on the leaves?
69 edge of hyphal -> rephrase
69-71 this sentence is not completely clear as worded
72 should -> were
73 graphs -> photos
76 where were they deposited?
79 Voucher specimens? Why not type material?
99 I don't understand how MEGA was used to process raw Sanger sequencing data
107-109 what partitioning scheme was used for the phylogenetic inference?
111 Partitions are mentioned but we don't know how the data were partitioned. Were introns partitioned separately from exons? Were the codon positions within the exons partitioned as well?
114-116 This description of a MrBayes analysis does not make any mention of how the runs were confirmed to have converged, a necessary step for any Bayesian inference. And 2,000,000 generations doesn't sound like enough for that to happen.
122-123 you have these listed out of order
136-138 Why not use a model test for the ML analysis?
164 trapezoidal
168 slightly bent
178 "independent clade" while this is technically not wrong, I believe it is somewhat misleading. I would omit it - it is simply enough to state which species it is close to.
183-185 sentence needs improvement
207 slightly bent
222 clavata
226-227 sentence needs improvement
285 this sentences has grammatical issues, but I also disagree with it scientifically. Narrowing of host range can take place within a genus, so what applies to one species of Pestalotiopsis doesn't necessarily hold true for the rest of the genus.
295-297 this is a great place to mention the importance of taxonomy and the deposition of specimens!
See my specific line comments for English issues.
Author Response
Dear sir/madam,
Thank you for your letter and comments concerning our manuscript “Morphological and phylogenetic analyses reveal three new species of Pestalotiopsis (Sporocadaceae, Amphisphaeriales) from Hainan, China” (Submission microorganisms-2427886).
Your comments are highly insightful and help us greatly improve the quality of our manuscript. We hope that the revisions and our responses as in the postscript would be sufficient to make our manuscript suitable for publication in Microorganisms.
Sincerely yours,
Zhaoxue Zhang

Round 2
Reviewer 2 Report
I suggest again moving Table 2 in the supplementary data. This would make the reading of the paper easier, and will also also look better. Why having that info spread over 4 pages? If somebody will be interested in the species used for comparative studies, he/she can go to the supplementary data is. However, not too many people will do it!
English got improved as the authors accepted the changes suggested by the reviewer.
Author Response
Dear sir/madam,
Thank you for your letter and comments concerning our manuscript “Morphological and phylogenetic analyses reveal three new species of Pestalotiopsis (Sporocadaceae, Amphisphaeriales) from Hainan, China” (microorganisms-2427886).
Your comments are highly insightful and help us greatly improve the quality of our manuscript. We hope that the revisions and our responses as in the postscript would be sufficient to make our manuscript suitable for publication in Microorganisms.
I have move Table 2 in the supplementary data.
Sincerely yours,
Zhaoxue Zhang